# Direct and indirect nutritional factors that determine reproductive performance of heifer and primiparous cows

**Lidiane R. Eloy**[1]*, **Carolina Bremm**[1], **José F. P. Lobato**[1], **Luciana Pötter**[2], **Emilio A. Laca**[3]

**1** Animal Science Department, Faculty of Agronomy, Federal University of Rio Grande do Sul, Porto Alegre, Rio Grande do Sul, Brazil, **2** Animal Science Department, Rural Science Center, Federal University of Santa Maria, Santa Maria, Rio Grande do Sul, Brazil, **3** Department of Plant Sciences, University of California, Davis, California, United States of America

* lidianeloy@hotmail.com

**Data Availability Statement:** Data are available within the Supporting Information files.

**Funding:** Funding for this research was provided by grants from the Brazilian National Council for

## Abstract

Pregnancy rate is a major determinant of population dynamics of wild ungulates and of productivity of livestock systems. Allocation of feeding resources, including stocking rates, prior to and during the breeding season is a crucial determinant of this vital rate. Thus, quantification of effects and interaction among multiple factors that affect pregnancy rate is essential for management and conservation of pasture-based systems. Pregnancy rate of 2982 heifers and primiparous cows was studied as a function of animal category, average daily gain during the breeding season, stocking rate, pasture type and body weight at the beginning of the breeding season. Data were obtained from 43 experiments conducted in commercial ranches and research stations in the Pampas region between 1976 and 2015. Stocking rate ranged from 200 to 464 kg live weight ha$^{-1}$, which brackets values for most of the grazing-lands in similar regions. Age at breeding was 14–36 months (24.6±7.5 months); initial breeding weights were 129–506 kg and 194–570 kg for heifers and primiparous cows. Pregnancy rate was modeled with an apriori set of explanatory variables where proximate variables (breed, body weight at start of breeding, weight gain during breeding and category) were included first and subsequently modeled as functions of other variables (pasture type, supplementation and stocking rate). This modeling approach allowed detection of direct and indirect effects (through nutrition and body weight) of factors that affect pregnancy rate. *Bos taurus* breeds (N = 1058) had higher pregnancy rate than *B. Taurus* x *B. indicus* crossbreed (N = 1924) females. Pregnancy rate of heifers and primiparous cows grazing in natural grasslands decreased with increasing stocking rate, but no effect of stocking rate was detected in cultivated and improved pastures. Pregnancy rate increased with increasing average daily gain during the breeding season. Use of cultivated or improved natural pastures promotes higher pregnancy rate, as well as allows an increase in stocking rate at the regional level. Body weight at the start of the breeding season is the primary determinant of pregnancy rates in heifer and primiparous cows.

Scientific and Technological Development (CNPq) and CAPES. The funders had no role in study design, data collection and analysis, decision to publish, or preparation of the manuscript.

**Competing interests:** The authors declare they have no conflict of interest.

## Introduction

Although fundamental cattle physiology and its relation to nutrition are well known, there is not sufficient quantitative information about how management affects nutrition and reproduction, especially under natural grasslands conditions. In particular, few studies have been done to evaluate the relative effects of stocking rate and other nutritional management factors on pregnancy rates at a regional level [1–6]. The lack of single studies with regional scope is understandable because such studies with livestock are logistically complex and extremely expensive.

An alternative to specific comprehensive studies is to analyze data pooled from multiple studies [7, 8] that address the same research question using equivalent response and explanatory variables, that is, a joint analysis of multiple experiments [9]. Such joint analysis requires care in the process of systematizing results from multiple studies but it has the advantage of increasing precision, decreasing costs and research time, and increasing the degrees of freedom in the analysis [10, 11].

Livestock production in the Pampas region of Southern Brazil is characterized by a low pregnancy rate that has remained stagnant over many years, despite multiple changes in economic and technological factors that affect productivity [12]. Low pregnancy rates prevent full development of the livestock sector in many regions of the world, and it may be associated with poor pasture management, overstocking and lack of differential nutritional management for animal categories with different requirements. Stocking rate is considered the most important decision in grazing management because it affects the forage base, herbage allowance, intake and animal performance [13]. Nutritional limitation during periods of high requirement can compromise development and delay puberty of heifers, as well as inhibit ovulation of cows [14].

Time at which puberty occurs relative to the start of breeding season is what determines pregnancy rate in the first breeding season of heifers [15], which influences a cow's ability to get pregnant in subsequent years and remain in the herd, determining her lifetime productivity. Puberty of heifers is influenced by management of the annual production cycle, as well as the physiology (production and release of hormones) and its genetic (breed and size of mature age) [16]. In addition to the use of pastures, body weight at the beginning of the breeding season is associated with animal nutrition and it is an important factor influencing the reproductive performance of heifers and beef cows [17–19]. Body condition score is a critical factor influencing nutritional status of beef cows and determining the success of artificial insemination [20].

Natural grasslands and cultivated pastures constitute the forage basis for beef cow herds in many regions of the world, including the US [21]. The grasslands that support cow-calf operation in the Pampas are characterized by spring-summer growth, with quality and availability reduced in autumn and winter. Cultivated and improved pastures are utilized to satisfy the nutritional requirements of cattle, especially during the cooler months when natural pasture growth is limited. Combined with cultivated and improved pastures, supplements may be used to increase average daily gain of grazing animals and to promote greater reproductive development [22].

The aim of the present study was to integrate information from multiple studies of factors that affect pregnancy rates in beef heifers and primiparous cows under production conditions in the Pampas to quantify response curves relating pregnancy rate to the most important predictors. First, we take an approach where pregnancy rate is analyzed as a function of known proximate factors such as initial body weight at breeding, category and average daily gains during the breeding season. Second, we add the effects of stocking rate, supplementation and

pasture type on proximate factors and directly on pregnancy rate to account for effects not mediated by the proximate factors evaluated. Our main hypothesis is that increases stocking rate will lead to reduced pregnancy rate, and that stocking rate effects on pregnancy rate are mediated by effects of stocking rate on body weight through weight gain during the breeding season. Furthermore, body weight at the beginning of the breeding season is expected to have a positive effect on pregnancy rate because of its relationship with physiological status and development. Primiparous cows are expected to have lower pregnancy than heifers due to the physiological stress imposed by recovery from pregnancy and lactation.

## Material and methods

### Ethics statement

All analyses were based on previously published studies: no ethics approval was required.

### Study sample

Data included records from 29 doctoral dissertations or master's theses for a total of 43 experiments (some studies had more than one experiment). Experiments were selected because the original raw data were available for all of them. Experiments were conducted at the Agronomic Experimental Station of Federal University of Rio Grande do Sul and in private ranches in Rio Grande do Sul, Southern Brazil, to investigate the effects of several factors on pregnancy rate of heifers and primiparous cows (Table 1) between 1976 and 2015. According to Köppen [23], climate in all sites represented in the data is subtropical humid.

### Data

The initial database created contained the following variables for each of 3933 animals (Table 2).

Breeds were recoded as crossbred (final N = 1924) vs. *B. Taurus* (final N = 1058). One experiment with an extreme stocking rate of 800.0 kg of body weight per hectare was excluded from the analysis. The remaining data had stocking rates ranging from 200 to 463.5 (average was 336.85±54.92) kg BW/ha, which are more typical for the region. Columns with large number of missing values and rows without complete multivariate records were excluded, resulting in a final data set of 2982 records (animals) for which all variables depicted in Fig 1 were available.

### Statistical analyses

Statistical analyses were conducted using R [53]. We used generalized linear mixed models (GLMM) based on the logit link function, as they are generally recommended for binary data [54, 55]. Data for each heifer and primiparous cow were available, allowing an analysis analogous to an incomplete block design, with a random intercept for each experiment [56].

The main response variable was pregnancy rate as evaluated by the relation between the number of pregnant heifers or primiparous cows and the total number of heifers or primiparous cows in each experiment. Explanatory factors considered were category, weight at the beginning of the breeding season, weight change during the breeding season, breed, stocking rate, and type of pasture before and after the breeding season. All quantitative variables were standardized to facilitate the convergence of the computations to estimate parameters. A structure of causal effects was established a priori (Fig 1) and then simplified by removing nonsignificant components.

**Table 1. Relation of studies of database with n (number of animals), location, coordinated geographic, precipitation and type of pasture in southern Brazil.**

| Author | n | Local | Coordinated geographic | Precipitation (mm/year) | Type of pasture* |
|---|---|---|---|---|---|
| Albospino, 1990 [24] | 23 | Eldorado do Sul | 30˚52'/51˚39' | 1332 | Italian ryegrass (*Lolium multiflorum* Lam.) |
| | | | | | Arrowleaf clover (*Trifolium vesiculosum*) |
| Azambuja, 2003 [25] | 216 | Arambaré | 31˚11'/51˚74' | - | Natural pasture |
| Beretta, 1994 [26] | 113 | Eldorado do Sul | 30˚52'/51˚39' | 1398 | Annual ryegrass (*Lolium multiflorum* Lam.) |
| | | | | | Arrowleaf clover (*Trifolium vesiculosum*) |
| Cachapuz, 1976 [27] | 57 | Dom Pedrito | 30˚99'/54˚70' | 1376 | Natural pasture |
| | | | | | Common vetch (*Vicia sativa*) |
| Deresz, 1976 [28] | 110 | Pelotas | 30˚58'/50˚40' | 1285 | Natural pasture |
| | | | | | Annual ryegrass (*Lolium multiflorum* Lam.) |
| | | | | | Arrowleaf clover (*Trifolium vesiculosum*) |
| Fagundes, 2001 [29] | 87 | Itaqui | 29˚24'/56˚47' | 1500 | Natural pasture |
| Freitas, 2005 [30] | 350 | São Gabriel | 30˚33'/54˚32' | 1193 | Annual ryegrass (*Lolium multiflorum* Lam.) White clover (*Trifolium repens*) |
| | | | | | Bird's-foot trefoil (*Lotus corniculatus*) |
| Gottschall, 1994 [31] | 114 | São Gabriel | 30˚33'/54˚32' | 1512 | Natural pasture |
| Lopes, 2004 [32] | 39 | Eldorado do Sul | 30˚52'/51˚39' | 1446 | Black oats (*Avena strigosa*) |
| | | | | | Annual ryegrass (*Lolium multiflorum* Lam.) Arrowleaf clover (*Trifolium vesiculosum*) |
| | | | | | Pearl millet (*Pennisetum americanum*) |
| Magalhães, 1992 [33] | 210 | Rosário do Sul | 30˚25'/54˚92' | 1550 | Annual ryegrass (*Lolium multiflorum* Lam.) |
| Marques, 2001 [34] | 231 | Eldorado do Sul | 30˚52'/51˚39' | 1440 | Black oats (*Avena strigosa*) |
| | | | | | Annual ryegrass (*Lolium multiflorum* Lam.) Arrowleaf clover (*Trifolium vesiculosum*) |
| Menegaz, 2006 [35] | 323 | Uruguaiana | 29˚76'/57˚09' | 1500 | Natural pasture |
| | | | | | Annual ryegrass (*Lolium multiflorum* Lam.) White clover (*Trifolium repens*) |
| | | | | | Bird's-foot trefoil (*Lotus corniculatus*) |
| Moraes, 1991 [36] | 60 | Dom Pedrito | 30˚99'/54˚70' | 1300 | Annual ryegrass (*Lolium multiflorum* Lam.) |
| | | | | | White clover (*Trifolium repens*) |
| | | | | | Sorghum (*Sorghum bicolor*) |
| | | | | | Bird's-foot trefoil (*Lotus corniculatus*) |
| Müller, 1998 [37] | 50 | Eldorado do Sul | 30˚52'/51˚39' | 1440 | Annual ryegrass (*Lolium multiflorum* Lam.) |
| | | | | | Arrowleaf clover (*Trifolium vesiculosum*) |
| Nardon, 1985 [38] | 65 | Eldorado do Sul | 30˚52'/51˚39' | 1398 | Annual ryegrass (*Lolium multiflorum* Lam.) |
| | | | | | Arrowleaf clover (*Trifolium vesiculosum*) |
| Pereira Neto, 1996 [39] | 62 | Eldorado do Sul | 30˚52'/51˚39' | 1332 | Annual ryegrass (*Lolium multiflorum* Lam.) |
| | | | | | Arrowleaf clover (*Trifolium vesiculosum*) |
| Pilau, 2007 [40] | 234 | Tupanciretã | 29˚03'/53˚48' | - | Black oats (*Avena strigosa*) |
| | | | | | Annual ryegrass (*Lolium multiflorum* Lam.) |
| Polli, 1986 [41] | 71 | Eldorado do Sul | 30˚52'/51˚39' | 1398 | Natural pasture |
| | | | | | Annual ryegrass (*Lolium multiflorum* Lam.) |
| | | | | | Arrowleaf clover (*Trifolium vesiculosum*) |
| Pötter, 2002 [42] | 92 | Quaraí | 30˚26'/56˚01' | 1356 | Natural pasture |
| | | | | | Annual ryegrass (*Lolium multiflorum* Lam.) |
| | | | | | Bird's-foot trefoil (*Lotus corniculatus*) |
| Quadros, 1991 [43] | 69 | Dom Pedrito | 30˚99'/54˚70' | 1540 | Natural pasture |

*(Continued)*

**Table 1.** (Continued)

| Author | n | Local | Coordinated geographic | Precipitation (mm/ year) | Type of pasture* |
|--------|---|-------|------------------------|--------------------------|------------------|
| Ribeiro, 1986 [44] | 70 | Cachoeira do Sul | 30˚03'/52˚89' | 1621 | Natural pasture |
| Rocha, 1997 [45] | 394 | Dom Pedrito | 30˚44'/54˚47' | 1450 | Annual ryegrass (*Lolium multiflorum* Lam.) |
| | | | | | White clover (*Trifolium repens*) |
| | | | | | Red clover (*Trifolium pratensis*) |
| | | | | | Bird's-foot trefoil (*Lotus corniculatus*) |
| Rosa, 2010 [46] | 241 | Dom Pedrito | 30˚44'/54˚47' | 1300 | Natural pasture |
| | | | | | Annual ryegrass (*Lolium multiflorum* Lam.) White clover (*Trifolium repens*) |
| | | | | | Bird's-foot trefoil (*Lotus corniculatus*) |
| Silva, 2010 [47] | 142 | Bagé | 31˚22'/54˚39' | 1300 | Annual ryegrass (*Lolium multiflorum* Lam.) |
| Simeone, 1995 [48] | 119 | Bagé | 31˚22'/54˚39' | 1350 | Natural pasture |
| Souza, 2005 [49] | 64 | Dom Pedrito | 30˚99'/54˚70' | 1376 | Annual ryegrass (*Lolium multiflorum* Lam.) |
| | | | | | White clover (*Trifolium repens* L.) |
| Souza, 2014 [50] | 49 | Júlio de Castilhos | 29˚23'/53˚68' | - | Black oats (*Avena strigosa* Schreb.) |
| | | | | | Palisade grass (*Urochloa brizantha*) |
| Tanure, 2008 [51] | 194 | Quaraí | 30˚26'/56˚01' | 1356 | Natural pasture |
| Zanotta Jr, 1984 [52] | 84 | Pelotas | 30˚58'/50˚40' | 1285 | Natural pasture |
| | | | | | Annual ryegrass (*Lolium multiflorum* Lam.) |
| | | | | | White clover (*Trifolium repens*) |

*Natural pasture with prevalence of Bahiagrass (*Paspalum notatum*), Dallisgrass (*Paspalum dilatatum*), *Axonopus affinis*, *Andropogon lateralis*, *Trifolium polymorphum*.

First, pregnancy rate (PR) was analyzed using *glmer* with a binomial distribution and a logit link, as a function of known proximate factors such as body weight and category. Breed was included to account for inherent differences in breeds that could modulate the effects of body weight, for example, due to differences in mature body or frame size. Model selection proceeded by simplification of a full model until it had only those effects that were significant or part of significant interactions. Significance of terms was assessed by type II Wald tests using the Anova() function of the *car* package [27]. The initial full model, expressed as an R formula for the generalized linear mixed-effects model (glmer) function of the lme4 package [57] was

$$\text{PR} \sim \text{breed} * \text{start.bw} + \text{categ} * \text{s.dwt} + \text{s.dwt} * \text{breed} + \text{s.dwt} * \text{start.bw} + \text{I(start.bw}^2) + \text{I(s.dwt}^2) + (1 \mid \text{experiment})$$

where *categ* is animal category, *start.bw* is weight at the beginning of the breeding season, *s. dwt* is daily weight gain, *I(start.bw^2)* is squared weight at the beginning of the breeding season, *I(s.dwt^2)* is daily weight gain squared and *experiment* is a categorical variable or factor with a different value for each experiment. Each experiment was allowed a random effect to account for the potential intraclass correlation caused by common condition for all animals in each experiment. The "*" operator indicates that both main effects and their interaction are included in the model. This final model after simplifications was tested against the full model by a likelihood-ratio test using the anova() function to make sure they were not significantly different.

Second, stocking rate was added as the last term to the resulting model to determine if stocking rate had significant effects beyond those effected via proximate factors. Significance of stocking rate effects was assessed with the same Wald test as before. Third, body weight at

**Table 2. Average and standard error of each variable in the initial database.**

| Variable | Average |
|---|---|
| Age at the beginning of the breeding season[1] | 24.6±7.5 months |
| Animal categories | |
| Heifers | 2257 females |
| Primiparous cows | 1676 females |
| Body weigth at the beginning of the breeding season | 315.4±55.9 kg |
| Body weigth at the end of the breeding season | 337.3±53.4 kg |
| Breeds | |
| Angus | 306 females |
| Braford | 499 females |
| Brangus | 323 females |
| Crossbred | 1928 females |
| Devon | 110 females |
| Hereford | 767 females |
| Body condition score at the beginning of the breeding season[2] | 3.2±0.6 |
| Body condition score at the end of the breeding season[2] | 3.4±0.6 |
| Stocking rate[3] | 337.32±54.68 kg BW/ha |
| Pasture types | |
| Cultivated | 2050 |
| Improved pasture[4] | 324 |
| Natural grassland | 1559 |
| Feed supplementation before the breeding season | |
| Not supplemented | - |
| Supplemented | |
| Brown rice bran | 0.5 to 1.0% BW |
| Commercial concentrate | 0.7 to 1.5% BW |
| Corn grain | 0.5% BW |
| Deffated rice bran | 1.5% BW |
| Deffated rice bran and sorgum silage | 1.5% BW |
| Ground corn grain | 0.7% BW |
| Sorghum silage and commercial concentrate | 1.5% BW |
| Protein salt | 0.1% BW |
| Rice and soy bran | 0.56% BW |
| Ryegrass and White clover hay | 0.28% BW |
| Sectaria hay | 0.92% BW |
| Sorghum silage and commercial concentrate | 1.5% BW |

[1] range 14 to 36 months;

[2] 0 to 5 scale;

[3] range 200 to 464 kg of body weight per hectare;

[4] Improved pasture were natural pastures with addition of fertilizer and seed of cultivated species in broadcast or sod-seeding applications.

the beginning of the breeding season was modeled with the following full model:

$$\text{start.bwt} \sim \text{categ} + \text{breed} + \text{pasture.pre} * \text{s.sr} + \text{pasture.pre} * I(\text{s.sr}^2) + \text{sup.pre} + (1 \mid \text{experiment})$$

where pasture.pre is a factor indicating whether animals grazed natural grassland, cultivated

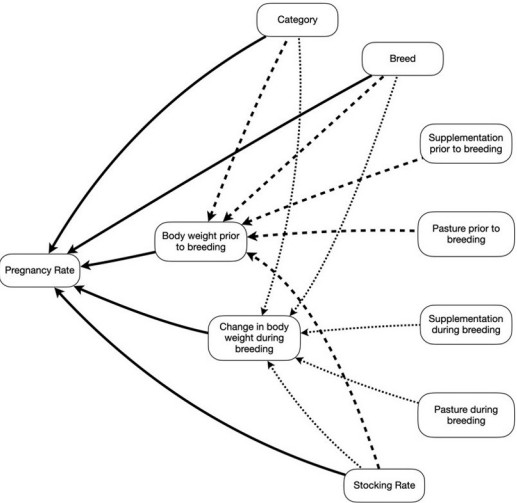

**Fig 1. Schematic structure of the hypothesized determinants of pregnancy rate (PR).** Full lines represent direct effects, dotted lines represent indirect effects through change in body weight during breeding, and dashed lines represent indirect effects through body weight prior to breeding.

pastures or improved grassland prior to the breeding period; s.sr is stocking rate, I(s.sr^2) is the quadratic effect of stocking rate, and sup.pre is a binary variable indicating whether animals received supplementation before of the breeding period. Other terms were defined above. Finally, change in body weight during the breeding period (s.dwt) was analyzed starting with the following full model:

$$\text{s.dwt} \sim \text{categ} + \text{breed} + \text{pasture.breed} + \text{s.sr} + \text{sup.breed} + I(.\text{s.sr}\,\hat{}\,2)$$
$$+ \text{pasture.breed} : \text{s.sr} + \text{categ} : \text{s.sr} + \text{breed} : \text{s.sr} + (1 \mid \text{experiment})$$

where pasture.breed is the type of pasture grazed during the breeding period and sup.breed is a binary variable indicating whether animals received supplementation during the breeding period. Models for start.bwt and s.dwt were simplified and final models were tested following the same procedures as before. For all models, assumptions were assessed by inspection of residual plots.

**Table 3. Analysis of variance of the model for pregnancy rate with proximate factors.**

| Effect | Wald's Chi-sq (type II) | df | p-value |
|---|---|---|---|
| start.bw[1] | 234.9 | 1 | <0.0001 |
| I(start.bw^2)[2] | 48.2 | 1 | <0.0001 |
| s.dwt[3] | 11.1 | 1 | 0.0009 |
| categ[4] | 10.7 | 1 | 0.0011 |
| start.bw:breed[5] | 6.2 | 1 | 0.0141 |
| breed[6] | 2.7 | 1 | 0.0995 |

[1]Weight at the beginning of the breeding period (standardized using mean = 318 kg, s = 69 kg).

[2]Quadratic effect of weight at the beginning of the breeding season.

[3]Average daily gain during the breeding season (standardized using mean = 0.264 kg/day, s = 0.314 kg/day).

[4]Animal category (heifer or primiparous cow).

[5]Interaction between weight at the beginning of the breeding season and breed.

[6] Breed type (crossbred or *B. taurus*).

## Results

### Effects of proximate causal factors on pregnancy rate

The final model for selected pregnancy rate was:

$$PR \sim \text{categ} + \text{start.bw} * \text{breed} + \text{s.dwt} + I(\text{start}\hat{}2) + (1 \mid \text{experiment}).$$

The most important factor affecting pregnancy rate was body weight at the beginning of the breeding season (Table 3), which accounted for 68% of the model sum of squares. Mc Fadden's pseudo $R^2$ [58] for fixed effects of the complete final model was 9.53%, whereas a model with only the linear and quadratic effects of initial body weight had a pseudo $R^2$ equal to 8.8%.

Pregnancy rate increased steeply with increasing body weight at the beginning of the breeding season for crossbreed and *B. taurus* females, but for *B. taurus* females it increased faster and reached a higher maximum than for crossbreed females. *B. taurus* females starting the breeding season with an average weight of 440.0 kg or more had an expected pregnancy rate of 99.0%, whereas crossbred females that started the breeding season with similar weight had an expected pregnancy rate of 91.0% (Fig 2).

Pregnancy rate increased with increasing average daily gain during the breeding period. Averaging over other predictors, the model estimated that pregnancy rate increases from 59% when animals lose 340 g per day to 79% when they gain 890 g per day during the breeding season. When daily gain was at its average (0.264±0.006 kg per day), pregnancy rate increased 1.7% per 100 g of daily gain during the breeding period. When all covariates are at their average values for both categories, pregnancy rate was 25 percentage points higher for heifers than for primiparous cows (80 vs. 55%).

### Effects of stocking rate not mediated by proximate factors

The addition of stocking rate as an explanatory factor resulted in the following model:

$$\text{preg} \sim \text{start.bw} + \text{s.dwt} + I(\text{start.bw}\hat{}2) + \text{breed} + \text{categ} + \text{s.sr}$$
$$+ \text{start.bw} : \text{breed} + \text{s.sr} : \text{start.bw} + (1 \mid \text{experiment}).$$

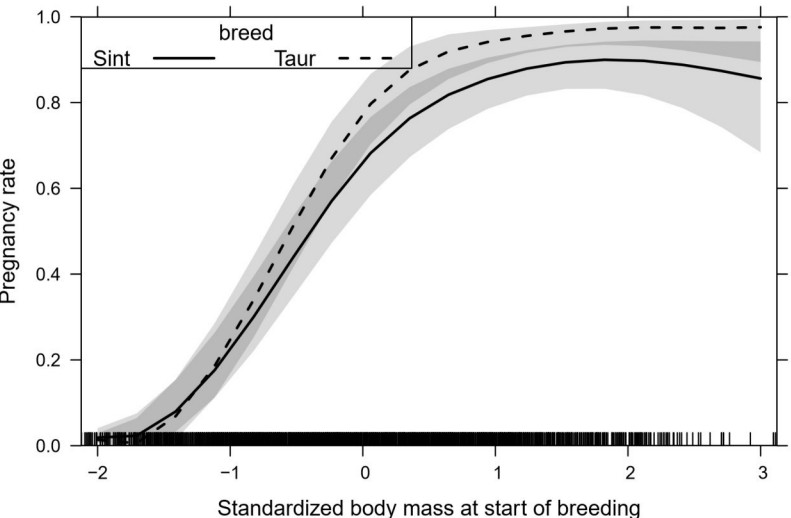

**Fig 2. Interaction between body weight at the beginning of the breeding season and Breed to pregnancy rate of heifers and primiparous cows (Sint: Crossbred, Taur: *B. taurus* females).** Shaded strips represent 95% confidence intervals for the expected value. Body mass average and standard deviation were 318 and 69 kg.

The contribution of stocking rate to explain variation in pregnancy rate was evaluated by entering stocking rate and its interactions last into the model and using a type II Wald test. Stocking rate and its interaction with body weight at the start of the breeding season contributed 4.7% of the total sum of squares of the model and had direct effects on pregnancy rate that were significant even after controlling for the potential indirect effects through proximate variables such as body weight and weight change during breeding (Table 4). Stocking rate exhibited a significant interaction with weight at the start of the breeding season by which the effect of stocking rate was small for body weight below the 318 kg average and negative for heavier animals (Fig 3).

Although stocking rate did have effects on pregnancy rate beyond those through proximate causal variables, the effects of other variables did not change much by the incorporation of stocking rate. The most important factor affecting pregnancy rate when stocking rate was added in the model continued to be body weight at the beginning of the breeding season, accounting for 84.4% of the sum of squares explained by the generalized mixed model (Table 4). The second largest contribution to the sum of squares of the model was due to average daily gain during the breeding season, which accounted for 4.3% of the explained variation (Table 4). The response to daily gain was similar to that in the model without stocking rate; pregnancy rate increased 1.5% per 100 g of daily gain when daily gain was at its average. The third largest contribution to the model's sum of squares was due to animal category, accounting for 3.0% of the explained variation. Pregnancy rate was greater in heifers than in primiparous cows (81 vs. 56%, p = 0.0054).

## Body weight at the beginning of the breeding season

Because supplementation did not have detectable effects on body weight at the start of the breeding season, the final model was

$$\text{start.breed.wt} \sim \text{categ} + \text{breed} + \text{pasture.pre} + \text{s.sr} + I(\text{s.sr}\string^2) +$$
$$\text{pasture.pre} : \text{s.sr} + \text{pasture.pre} : I(\text{s.sr}\string^2) + (1 \mid \text{experiment}),$$

where *start.breed.wt* is body weight in kg at the start of the breeding season and *pasture.pre* is the type of pasture grazed prior to the breeding season. Thirty nine percent of the variation in initial body weight was explained by the fixed effects of the model, and an additional 41% was explained by variation among experiments (random effects variation due to differences between experiments in variables not measured). Pasture type accounted for 60% of the sum

**Table 4. Analysis of variance of the pregnancy rate model with proximate factors and stocking rate.**

| Variables | Wald's Chi-sq (type II) | df | p-value |
|---|---|---|---|
| start.bw | 227.4 | 1 | <0.0001 |
| I(start.bw^2) | 44.1 | 1 | <0.0001 |
| s.dwt | 9.5 | 1 | 0.0021 |
| Categ | 7.7 | 1 | 0.0054 |
| start.bw:breed | 7.1 | 1 | 0.0076 |
| s.sr[1] | 6.1 | 1 | 0.0135 |
| start.bw:s.sr[2] | 6.0 | 1 | 0.0144 |
| Breed | 2.9 | 1 | 0.0883 |

[1]Stocking rate (standardized using mean = 328 kg/ha and s = 58.9 kg/ha).

[2]Interaction between weight at the start of the breeding season and stocking rate.

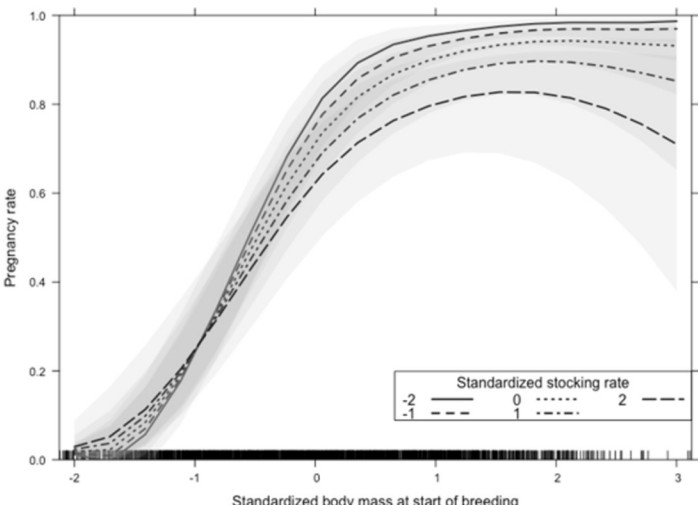

**Fig 3. Interaction between body weight at the beginning of the breeding season and stocking rate on pregnancy rate of heifers and primiparous cows.** Shaded areas are 95% confidence intervals. Tickmarks above the horizontal axis represent observations.

of squares explained by fixed effects, that is, 0.6 * 39% = 23.4% of the total variation in initial body weight. The interaction between pasture type and stocking rate accounted for 27.9% of the sum of squares explained by fixed effects, that is, 0.279 * 39% = 10.9% of the total variation in initial body weight. Body weight declined quadratically with increasing stocking rate in natural pastures, but it was not affected by the range of stocking rates studied in cultivated or improved pastures (Fig 4). At the average stocking rate of 328 kg/ha, initial body weight of animals grazing cultivated and improved pastures was 15 kg greater than that of animals grazing natural pastures, and this difference increased to 44 kg when stocking rate increased to 388 kg/ha.

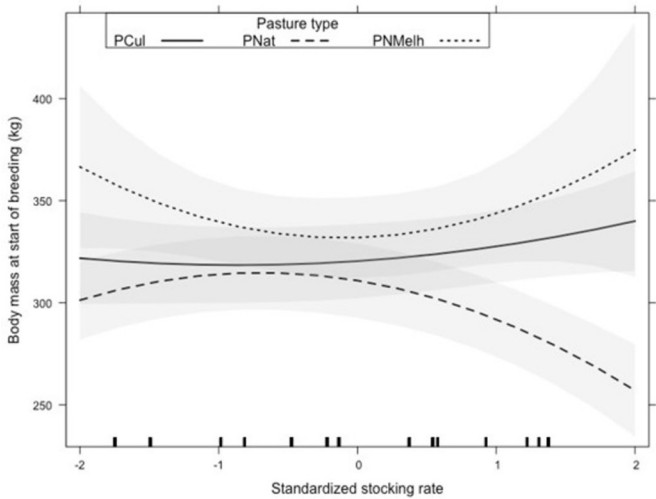

**Fig 4. Effects of stocking rate and type of pasture on body weight at the beginning of the breeding season of heifers and primiparous cows.** Pcul: cultivated pastures; PNMelh: improved pastures; PNat: native grasslands. P-value refer to the hypotheses that stocking rate has no effect on body weight at the start of the breeding season. Tick marks immediately above the X axis indicate the values of stocking rate present in the sample.

Breed type and category had effects on initial weight that were independent of stocking rate. Ninety five percent confidence intervals for weight at the start of the breeding season were 303 to 338 and 295 to 330 kg for crossbred and *B. taurus* types. Confidence intervals for heifers and primiparous cows were 263 to 313 and 332 to 380 kg.

## Change in body weight during the breeding season

The final model for average daily gain during the breeding season was

$$\text{s.dwt} \sim \text{pasture.breed} + \text{s.sr} + I(\text{s.sr}^{\wedge}2) + (1 \mid \text{experiment}),$$

where *pasture.breed* is the type of pasture grazed during the breeding season.

Stocking rate and type of pasture grazed accounted for equal parts of the total variarion explained by the fixed effects of the model. Average daily gain during the breeding season decreased quadratically with increasing stocking rate (Fig 5).

## Discussion

This study is unique because it integrated data from thousands of individual animals from multiple sites and experiments and because it established quantitative relationships between reproductive performance and stocking rate. Although stocking rate is recognized as one of the most important factors determining productivity of grazing systems [59], studies relating grazing animal performance to stocking rate are rare. A Web of Science search performed on 22 July 2021 with the terms "beef cattle" AND "pregnancy rate" AND "stocking rate" yielded nine articles, only one of which [60] presented original data on the effects of stocking rate on pregnancy rates. Most of the studies where stocking rate is considered as one of the explanatory variables for animal performance include few levels of stocking rate that explore a very limited range. We surmise that as a consequence of the limited range explored and the inherent high variability of herd-level studies, many studies failed to detect effects of stocking rate. The present study included 16 levels of stocking rate ranging from 0.5 to 1.2 head/ha or 200 to 464 kg/ha, which allowed quantification of response curves.

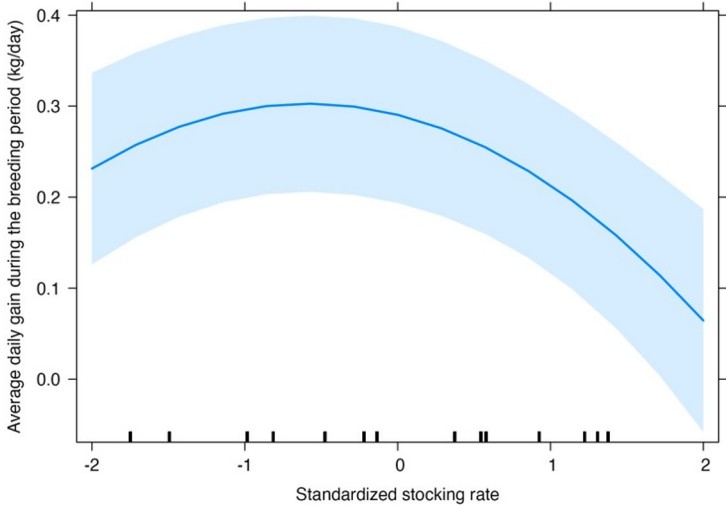

**Fig 5. Average daily gain during the breeding season and stocking rate of heifers and primiparous cows.** Shaded strip shows the 95% confidence band.

Although body weight at the beginning of the breeding season and average daily gains during the breeding season were dominant in explaining pregnancy rate in the experiments analyzed, multiple factors determine pregnancy rate, many of which are not closely related to body weight, animal category or breed. Other factors explored in some of the experiments analyzed, such as weaning method and use of artificial insemination, were considered and did not have detectable effects in preliminary models for pregnancy rate. However, the different weaning methods and use of artificial insemination were not represented across a good range of values in the other factors.

## Effects of proximate causal factors on pregnancy rate

The effects of body weight at the beginning of the breeding season, animal category and changes in body weight during the breeding season were quantified and yielded response curves with low variance. For example, the pseudo coefficient of variation (CI half width/(2 expected value)) of pregnancy rate for *B. taurus* cows at average weight at the beginning of the breeding season was 6.5%. The most important causal factor influencing pregnancy rate was body weight at the beginning of the breeding season, which interacted with breed to determine that at high initial weights, *B. taurus* females had higher pregnancy rate than crossbred females (Fig 2). The higher pregnancy rate observed in *B. taurus* cows may be due to the higher selection experienced by these females or to the smaller size and lower milk production of those relative to crossbreed cows, which have higher weight and milk production [61]. Pregnancy rate is influenced by nutrition, because it directly affects the reproductive physiology in beef cows [6], mainly in periods of higher requirements like pre and postpartum. If nutrition is inadequate, body reserves become depleted and body condition declines [62], resulting in low ovulation rate. Females with adequate metabolic status and high body weight have high levels of glucose, insulin and growth factor I (IGF-I) [63, 64], potentiating the effect of gonadotrophins (LH and FSH) [65] and promoting ovulation [66]. Pre and postpartum periods coincide with low availability of nutrients in natural grasslands, which are characterized by variation in composition, structure and, seasonality of production and quality [67, 68].

Animals with smaller frame reach physiological maturity earlier, at a lower weight and with greater fat content than larger animals [69]. When growth rate decreases and the process of fat deposition begins, larger animals are still in the growth phase [70]. In addition, the higher pregnancy rate observed in the *B. taurus* females can be explained by the greater selection for precocity carried out in the herds from which these females proceed [62]. When heifers reach puberty and mate earlier the biological efficiency of the herd is improved for as long as the early mating does not compromise full development. These characteristics become more important as production systems become more intensive and competitive. Reducing the age at first conception alters the structure of the herd and shortens the interval between generations, thus decreasing the participation of unproductive animals in the composition of the herd [71, 72].

Weight gain during the breeding season is clearly important for cows to become pregnant. Greater weight gain in this period indicates that forage is less limiting, and that sufficient quantity and quality of food intake is obtained to support ovarian activity [73]. According to [74], there is greater biological efficacy in females that have their first calf at about two, rather than three or more years of age. Adequate weights at the beginning of the breeding season are decisive for a high conception rate [75].

The lower pregnancy rate of primiparous cows than heifers may be related to the stress of calving and the combined effects of growth and first lactation requirements of primiparous cows. Low reproductive success has been documented for primiparous animals when they are subjected to periods of pre or postpartum feeding restriction [76]. The negative direct effect of

primiparous condition on pregnancy rate appeared with the inclusion of initial body weight, but primiparous condition had an indirect positive effect on pregnancy rate relative to heifers through the fact that they were heavier than heifers at the beginning of the mating period (Fig 6).

Stocking rate interacted with body weight at the beginning of the breeding season, whereby higher stocking rates were associated with lower pregnancy rate only in the high range of body weight (Fig 3). Lower stocking rates allow greater development of the animal, due to the higher forage accumulation, making it possible for females to have more food available [77]. Stocking rate is a primary management variable in grazing systems because it modulates the interactions between animals and pasture [78, 79]. As stocking rate increases, herbage allowance decreases and can reach levels where intake per animal is too low for production, but intake per unit area surpasses the ability of pastures to produce and recover. Individual animal performance decreases as stocking rate increases, because the daily intake is constrained by limiting sward structure at low herbage allowance [80], but production per unit area increases and then decreases with increasing stocking rate [67, 68, 81]. High stocking rates pre and post partum make it difficult for cows to recover good body condition after calving, compromising the reproductive performance of the cow and the productivity in subsequent seasons and

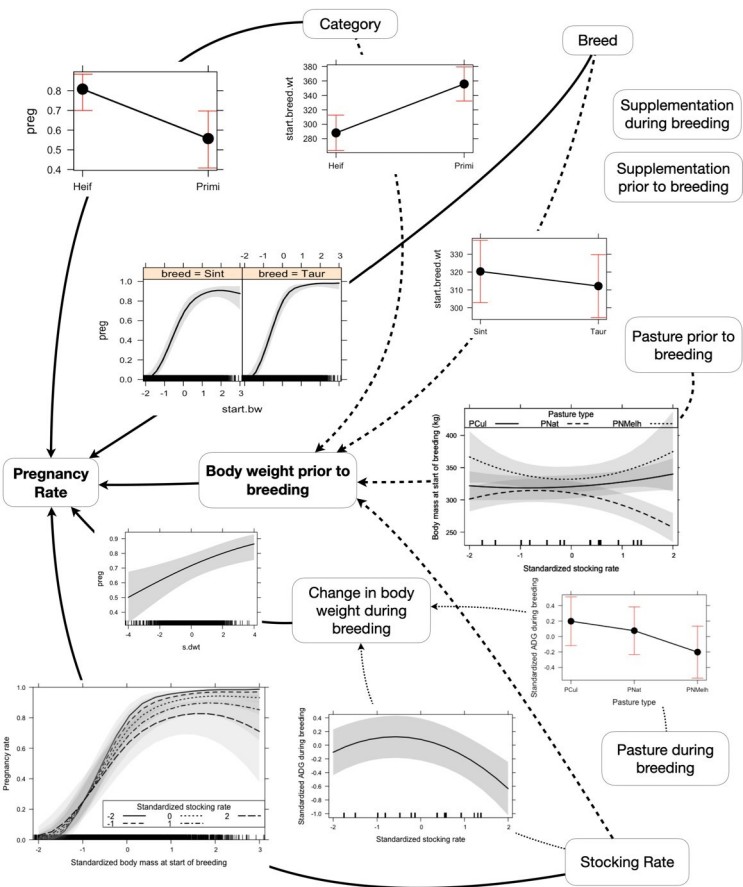

**Fig 6. Schematic representation of modeling results showing effects of animal factors (category and breed type), foraging environment (pasture type, stocking rate and supplementation) on body weight and reproductive performance of beef cattle.** Full lines represent direct effects, dotted lines represent indirect effects through change in body weight during breeding, and dashed lines represent indirect effects through body weight prior to breeding. Absence of a line indicates the factor was not included in the final model.

reproductive years [82]. Therefore, stocking decisions must be informed by curves that relate individual performance to stocking rate like the ones provided in the present study. These curves are particularly important for the integration of biological and economic functions to determine optimal stocking rates.

As expected, heifers achieve greater pregnancy rates when maintained in good nutritional conditions. Inadequate management practices, such as excessive stocking rate and lack of custom management for certain animal categories have led to generally low indices of productivity in the region. However, there are possibilities for reducing the age of slaughter and the age at first breeding, which may allow the improvement of productive and reproductive indices [83, 84]. Because natural pastures of the region are dominated by warm season grasses with low productivity and quality in the cold season, grazing of cultivated cool-season pastures significantly increases indicators of reproductive performance, beef yield and economic results [85].

In agreement with [86], we observed that when heifers and primiparous cows are maintained in optimal conditions of grazing and nutrition, that is, maintained in high quality pastures, with sufficient body weight and intermediate stocking rates, they achieve near maximal reproductive success. Sufficient nutrition allows early breeding, which increases the overall efficiency of production for the herd.

### Effects of stocking rate not mediated by proximate factors

Our results show that stocking rate has an effect on pregnancy rate that is not explained by any of the other variables considered. Even after controlling for effects of starting body weight and weight change during the breeding season, when body weight at the beginning of the breeding season is greater than average, pregnancy rate declines with increasing stocking rate. This effect of stocking rate appears to be restricted to the range of starting body weight where pregnancy rate no longer responds to body weight. This further suggests that stocking rate had an effect that was not mediated by the observed effects of stocking rate on body weight at the start of the breeding period and weight change during breeding (Fig 6).

Effects of stocking rate on pregnancy rate that are not related to nutritional condition, as reflected in body weight and daily gain, might be related to animal health and associated management variables. Higher stocking rates may result in greater load of external and internal parasites [87]. Tick infestation is common in this region, and ticks frequently carry Babesia [88]. When grazing at higher stocking rates, animals are forced to graze closer to the soil and increase the rate of ingestion of parasite helminth larvae [89]. However, herd health, particularly related to infections that directly or indirectly compromise the reproductive tract of females and the embryo and/or fetus, also stands out as an important factor of interference in the reproductive efficiency of beef cattle herds. In free herds, the introduction of the etiologic agent will cause, in most cases, various clinical signs such as repeated estrus, abortion, stillbirth, birth of weak animals and infertility [90]. Stocking rate might also affect pregnancy rate related to social interactions among bulls and cows [91].

### Body weight at the start of breeding season and changes in body weight during the breeding season

Our results agree with the conventional wisdom that stocking rate is one of the most important factors in grazing management. Stocking rate interacting with type of pasture before mating was the most important factor influencing body weight at the beginning of the breeding season (Fig 4) and it was the most important factor affecting the changes in body weight during the breeding season (Fig 5). The lowest weights at the beginning of the breeding season observed in this study are close to the minimum weight recommended for the first breeding season (50–

57% of adult weight) to avoid impairment of life-long reproductive performance [92, 93]. Body weight of heifers and cows grazing cultivated or improved pastures was high and did not respond to stocking rate, presumably because the level of feeding and pasture production was sufficient to provide enough nutrition at all stocking rates studies. On the other hand, we observed a typical response of declining in body weight and average daily gain during the breeding season for animals grazing native pastures where forage amount and quality are lower than in cultivated and improved pastures. Lower stocking rates allow animals to select better-quality diets, while higher stocking rates reduce vegetation abundance, constraining daily intake [80].

## Conclusions

This joint analysis of a large number of experiments conducted over decades in the Pampas region confirms the importance of body weight at the start of the breeding season to achieve high pregnancy rates in cattle. Because of the long term, large geographic region and large number of cows involved in this synthesis, results should be useful not only for ranch-level management but also for regional agricultural policy. Body weight at the beginning of the breeding season is an easily measurable variable and can be used as a herd reproductive management tool. Stocking rate had a negative effect on pregnancy rate both through its negative effects on initial body weight and weight change during the breeding season, and its direct negative effects on pregnancy rates when body weight was not limiting. The negative effect of stocking rate through body weight and weight gain supports our hypothesis that increases in stocking rate will lead to reduced pregnancy rate, and that stocking rate effects on pregnancy rate are mediated by body weight and weight gain. However, the presence of a direct effect of stocking rate not mediated by body weight or weight gain is strong evidence against the hypothesis that all effects were mediated by body weight. The mechanisms by which stocking rate affects pregnancy rate independently of body weight need further elucidation. Heifers tended to have lower body weight than primiparous cows, but after correction for body weight, they had greater pregnancy rates than primiparous cows, most likely due to the fact that primiparous cows were simultaneously lactating and growing. Response curves derived from our study can be used to optimize stocking rates under various economic conditions and to guide policies to improve the efficiency of reproductive livestock herds under free-grazing conditions.

## Supporting information

**S1 Data.**
(XLSX)

## Author Contributions

**Conceptualization:** Lidiane R. Eloy, Luciana Pötter, Emilio A. Laca.

**Data curation:** Lidiane R. Eloy, Emilio A. Laca.

**Formal analysis:** Lidiane R. Eloy, Emilio A. Laca.

**Funding acquisition:** Lidiane R. Eloy.

**Investigation:** Lidiane R. Eloy.

**Methodology:** Lidiane R. Eloy, Emilio A. Laca.

**Project administration:** Lidiane R. Eloy.

**Resources:** Lidiane R. Eloy.

**Supervision:** Carolina Bremm, José F. P. Lobato, Emilio A. Laca.

**Validation:** Emilio A. Laca.

**Visualization:** Lidiane R. Eloy, Emilio A. Laca.

**Writing – original draft:** Lidiane R. Eloy.

**Writing – review & editing:** Lidiane R. Eloy, Carolina Bremm, Emilio A. Laca.

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
