## [Decision Letter · Decision Letter 0]

7 Feb 2022

PONE-D-21-30153Direct and indirect nutritional factors that determine reproductive performance of heifer and primiparous cowsPLOS ONE

Dear Dr. Eloy,

Thank you for submitting your manuscript to PLOS ONE. After careful consideration, we feel that it has merit but does not fully meet PLOS ONE’s publication criteria as it currently stands. Therefore, we invite you to submit a revised version of the manuscript that addresses the points raised during the review process.

Dear author, please proceed to make the edits suggested by the reviewers and submit a revised version of your manuscript. I am including some of my suggestions as well in the attached document.

We look forward to receiving your revised manuscript.

Kind regards,

Luis Alonso Villalobos

Academic Editor

PLOS ONE

Journal Requirements:

2. PLOS ONE publication criterion #3 requires that the research must be described in enough detail to allow readers to fully replicate the study (https://journals.plos.org/plosone/s/criteria-for-publication#loc-3). Please could you provide more details on the following items:

a. The process of selecting studies (i.e. screening, eligibility)

b. Specify study characteristics used as criteria for eligibility, giving rationale for inclusion in this study.

c. For studies accessed remotely the full electronic search strategy for at least one database, including any limits used, such that it could be repeated. 

d. For studies that are unpublished, where the data may be available or if it is being published in this manuscript.

Funding for this research was provided by grants from the Brazilian National Council

for Scientific and Technological Development (CNPq) and CAPES.

No authors have competing interests.

Additional Editor Comments:

Dear author

We have received the comments from three reviewers. Two of them considered that your manuscript requires a minor revision and the third one suggested to reject it. In my role as Academic Editor I agree with the two reviewers and I consider that the manuscript has technical and scientific merit to be published on PLOS ONE.

Please read carefully the suggestions given by the reviewers and submit a revised version.

Reviewers' comments:

Reviewer's Responses to Questions

**Comments to the Author**

1. Is the manuscript technically sound, and do the data support the conclusions?

Reviewer #1: Yes

Reviewer #2: Partly

Reviewer #3: Yes

2. Has the statistical analysis been performed appropriately and rigorously? 

Reviewer #1: I Don't Know

Reviewer #2: Yes

Reviewer #3: Yes

3. Have the authors made all data underlying the findings in their manuscript fully available?

Reviewer #1: Yes

Reviewer #2: No

Reviewer #3: Yes

4. Is the manuscript presented in an intelligible fashion and written in standard English?

Reviewer #1: Yes

Reviewer #2: Yes

Reviewer #3: Yes

5. Review Comments to the Author

Reviewer #1: General Comment: The authors have used a vast amount of published information for analysis, and done so in a useful and understandable manner. I commend the authors on the extensive review and use of existing literature as citations in the manuscript. This increases the understanding and validity of the current research presented.

Statistical Analysis: As a reviewer I confess that I do not feel qualified to critique and fully understand the specifics and depth of statistical analysis. However, I trust that the approach of the authors is consistent will acceptable procedures.

Line by line suggestions to authors:

Abstract: In the 2nd to last sentence, the phrase "...as well as allows higher in stocking rate..." substitute "an increase" for "higher"

100-108: The authors do a good job of outlining the aim (objective) of the study, along with the approach to investigate their aim. However, no direct hypothesis is stated. I suggest the authors include a specific hypothesis in this section, and refer to and substantiate this hypothesis in the conclusion of the manuscript.

Table 1: This is an extensive and useful table to fully demonstrate the extent and potential application of the results across a wide expanse of conditions.

141: Provide a brief, yet informative statement to better define the term "improved"

143: Provide greater detail concerning the type, level and range of supplementation used.

231-232: The result of taurine (straight-bred) pregnancy rate increasing faster and higher maximum than crossbreed females is inconsistent with numerous evidence showing the impact of hybrid vigor on reproductive traits. Authors should address this finding. This result is also noted in lines 356-357, and referenced in Figure 2. In line 373, selection for precocity is a cause for this result, however this appears to be conjecture without evidence.

450: "weigh" should be corrected to "weight"

452-458: Could the decrease in pregnancy rate also be attributed to simply the stress of increased crowding and grazing competition of the animals at higher stocking rates?

484: suggest "use as a reproductive management tool"

491: correct "out" to "our"

656: Gregory is misspelled

Figures: My general comment is that these are helpful and prescriptive, however I offer a few suggestions;

Fig. 3 - horizontal is misspelled

Fig. 4 - is it possible to include the level of significance (i.e. P value) directly the figure?

Fig. 5 - similar to comment above, include evidence of quadratic decrease as stated in line 327 with the P value or similar notation directly in the figure. Also, suggest describing what the tick marks on the X axis are.

Fig. 6 - An illustrative representation of factors influencing pregnancy rate. I suggest that more visual contrast in the thickness of the lines (thickness) would improve the clarity of the diagram.

I thank the authors for undertaking this project. It was informative and expansive.

Reviewer #2: The rationale of the manuscript is very interesting and aims to clarify some aspects of cattle breeding even if the analysis is based on manuscript not available on bibliographic platforms. Furthermore, it is my opinion that the number of theses and subjects analyzed are not sufficient to justify the conclusions. For these reasons I reject the manuscript

Reviewer #3: This is a well done meta analysis, it should be useful for decision making in rangeland/pasture systems. Please consider these small edits.

on line 231 the phrase 'taurine females' is used and this is new to me, I have never seen this before. If I had to guess it means 'purebred.' but either a definition should be given or please use a different word.

line 339 cattle is misspelled

line 481, perhaps the word 'analysis' is more correct than 'synthesis' in this context,

6. PLOS authors have the option to publish the peer review history of their article (what does this mean?). If published, this will include your full peer review and any attached files.

Reviewer #1: No

Reviewer #2: No

Reviewer #3: **Yes: **John P McNamara

---

## [Author Response · Author response to Decision Letter 0]

1 Apr 2022

Dear Academic Editor and Reviewers,

The authors appreciate the comments of the editor and the reviewers, and have accepted the suggestions, which contributed to the improvement of the manuscript.

---

## [Decision Letter · Decision Letter 1]

20 Jun 2022

PONE-D-21-30153R1Direct and indirect nutritional factors that determine reproductive performance of heifer and primiparous cows1PLOS ONE

Dear Dr. Eloy,

Thank you for submitting your manuscript to PLOS ONE. After careful consideration, we feel that it has merit but does not fully meet PLOS ONE’s publication criteria as it currently stands. Therefore, we invite you to submit a revised version of the manuscript that addresses the points raised during the review process.

We look forward to receiving your revised manuscript.

Kind regards,

Luis Alonso Villalobos

Academic Editor

PLOS ONE

Journal Requirements:

Additional Editor Comments (if provided):

Dear author, we have received the comments of your manuscript and I am glad to inform you that a minor revision has been requested. Please go through the final edits suggested by the reviewers and send the second revised version so we can expedite the publication process. Let us know if you have any questions regarding the edits suggested or the process itself.

Reviewers' comments:

Reviewer's Responses to Questions

**Comments to the Author**

1. If the authors have adequately addressed your comments raised in a previous round of review and you feel that this manuscript is now acceptable for publication, you may indicate that here to bypass the “Comments to the Author” section, enter your conflict of interest statement in the “Confidential to Editor” section, and submit your "Accept" recommendation.

Reviewer #3: All comments have been addressed

Reviewer #4: (No Response)

Reviewer #5: (No Response)

2. Is the manuscript technically sound, and do the data support the conclusions?

Reviewer #3: Yes

Reviewer #4: Yes

Reviewer #5: Yes

3. Has the statistical analysis been performed appropriately and rigorously? 

Reviewer #3: Yes

Reviewer #4: Yes

Reviewer #5: Yes

4. Have the authors made all data underlying the findings in their manuscript fully available?

Reviewer #3: Yes

Reviewer #4: Yes

Reviewer #5: Yes

5. Is the manuscript presented in an intelligible fashion and written in standard English?

Reviewer #3: Yes

Reviewer #4: Yes

Reviewer #5: Yes

6. Review Comments to the Author

Reviewer #3: Thanks for a nice piece of work.

I have no idea why there is a minimum character count on this box that makes no sense to me, if I can say something in 30 characters why do I need to come up 70 more?

Reviewer #4: In general, the MS is well written, materials and methods section are clearly stated, as well as the results, discussion and main conclusions. Authors did a good job addressing all the previous comments, which improved the original MS. There are still some minor corrections, specially regarding the objective and hypothesis of the study.

Lines 99-101: It is necessary to include “in beef heifers and primiparous cows” in the objective.

Lines 103-104: I think it would be better to use “initial body weight at breeding” and “average daily gains during the breeding season” instead of “body weight” and “changes in body weight per day during breeding season”, respectively.

Lines 106-109: In the hypothesis there´s no mention to factors like: animal category, type of pasture, nor body weight at the beginning of the breeding season. What did the authors expected relative to those factors?

Lines 109-111: I think this would be more adequate for the discussion or conclusion section.

Line 235: change “with an average weight of 440.0 kg or more and had an expected pregnancy rate of 99.0%” to “with an average weight of 440.0 kg or more had an expected pregnancy rate of 99.0%”

Lines 319-324: This corresponds to the discussion section.

Lines 468-470: A reference for this would be necessary.

Reviewer #5: General: The study evaluates the factors determining the reproductive performance of beef heifers and cows in the Pampas Region. The study has a systemic approach and provides quantitative relationships which are considerably useful, and can be used to optimize stocking rates to improve heifers and cows reproductive performance.

The minor revisions I have are listed below:

62 From instead of form.

87 Mention the effects of body condition score on pregnancy rate.

99 Which are your hypothesis? I recommend to avoid using subjective adverbs such as “carefully”. I assume that if your objective is to quantify, you will do it carefully.

123 Did you considered the changes in breeds genetics during the period evaluated?

142 Which were the types of supplements used? Was there a range of supplementation levels?

187 Maybe the sentence: “This final model…” sould be change with “the final model after simplifications”, since it may generate confusions with the full model.

196 sup.pre was not defined.

202 sup.breed was not defined.

205 Is sup.pre defined correctly? Or was it confused with sup.breed?

Statistical Analyses If you reported de means of body condition score of your database, why didn´t you include it in your models?

233 “… 440.0 kg or more and had…” delete the “and”.

237 Interaction instead of Interactin

282 Instead of “near its average” I would suggest to use a range of daily gain.

360 Can you discuss this with your results of body condition score?

369 Couldn´t this be corrected by using the body condition score instead of the body weight in the models?

393-396 Couldn´t this be corrected by using the body condition score instead of the body weight in the models?

436 Correct the citation to “In agreement with Rovira (1974)”

452-458 It´s very interesting the analysis of the effects of stocking rate not mediated by the effects on BW at breeding and BW gain during breeding. You discussed that this effect occurs only when BW at the beginning of the breeding season is greater than average, and mentioned that parasites could be a possible cause of this response. However, parasites mainly affect weight gain. I suggest to discuss which would be the health and management variables that don´t affect the nutritional condition but do affect the reproductive performance.

484 be used IN herd reproductive management.

Figure 3 I would suggest to include body mass average and standard deviation as in Figure 2

Figure 4 and 5 I would suggest to include Stocking rate average and standard deviation as in Figure 2

Figure 5 The y axis title is incomplete.

Figure 6 This is a great representation, however the size of the figures is too small and it´s difficult to read. The references to the type of line are not clear, and it is hard to distinguish the thick of the lines.

7. PLOS authors have the option to publish the peer review history of their article (what does this mean?). If published, this will include your full peer review and any attached files.

Reviewer #3: **Yes: **John P McNamara

Reviewer #4: No

Reviewer #5: No

---

## [Author Response · Author response to Decision Letter 1]

19 Aug 2022

Dear Reviewers,

The authors appreciate the comments of the reviewers, and have accepted the suggestions, which contributed to the improvement of the manuscript. 

Changes were made using 'track changes' in MS Word document. Please find below the specific answers to the editor and reviewers’ comments.

Responses to the Reviewer 4 comments

Lines 99-101: It is necessary to include “in beef heifers and primiparous cows” in the objective.

Response: The sentence was added and the text was changed to:

The aim of the present study was to integrate information from multiple studies of factors that affect pregnancy rates in beef heifers and primiparous cows under production conditions in the Pampas to carefully quantify response curves relating pregnancy rate to the most important predictors.

Lines 103-104: I think it would be better to use “initial body weight at breeding” and “average daily gains during the breeding season” instead of “body weight” and “changes in body weight per day during breeding season”, respectively.

Response: The sentence was added and the text was changed to:

First, we take an approach where pregnancy rate is analyzed as a function of known proximate factors such as initial body weight at breeding, category and average daily gains during the breeding season.

Lines 106-109: In the hypothesis there’s no mention to factors like: animal category, type of pasture, nor body weight at the beginning of the breeding season. What did the authors expected relative to those factors?

Response: We added the following sentences to the discussion:

Furthermore, body weight at the beginning of the breeding season is expected to have a positive effect on pregnancy rate because of its relationship with physiological status and development. Primiparous cows are expected to have lower pregnancy than heifers due to the physiological stress imposed by recovery from pregnancy and lactation.

Lines 109-111: I think this would be more adequate for the discussion or conclusion section.

Response: The sentence was moved to the Conclusions section, after the first sentence therein.

Line 235: change “with an average weight of 440.0 kg or more and had an expected pregnancy rate of 99.0%” to “with an average weight of 440.0 kg or more had an expected pregnancy rate of 99.0%”.

Response: The word “and” was excluded. The sentence was changed to:

B. taurus females starting the breeding season with an average weight of 440.0 kg or more had an expected pregnancy rate of 99.0%, whereas crossbred females that started the breeding season with similar weight had an expected pregnancy rate of 91.0% (Fig 2). 

Lines 319-324: This corresponds to the discussion section.

Response: The paragraph was modified as shown below and moved to become the second paragraph in the discussion.

Although body weight at the beginning of the breeding season and average daily gains during the breeding season were dominant in explaining pregnancy rate in the experiments analyzed, multiple factors determine pregnancy rate, many of which are not closely related to body weight, animal category or breed. Other factors explored in some of the experiments analyzed, such as weaning method and use of artificial insemination, were considered and did not have detectable effects in preliminary models for pregnancy rate. However, the different weaning methods and use of artificial insemination were not represented across a good range of values in the other factors.

Lines 468-470: A reference for this would be necessary.

Response: The last sentence was modified and references were added. The last two sentences of the paragraph now read:

When grazing at higher stocking rates, animals are forced to graze closer to the soil and increase the rate of ingestion of parasite helminth larvae (Bransby 1993). Stocking rate might also affect pregnancy rate in ways unrelated to nutrition but linked to the effects on sources of stress related to social interactions among bulls and cows (Fernandez-Novo et al. 2020.

Responses to the Reviewer 5 comments

Line 62: From instead of form.

Response: The word “form” was changed to “from”. The phrase was re-written as follows:

An alternative to specific comprehensive studies is to analyze data pooled from multiple studies (Duffield, Merril & Bagg 2012; Lean, Thompson & Dunshea 2014) that address the same research question using equivalent response and explanatory variables, that is, a joint analysis of multiple experiments (Schwarzer, Carpenter & Rücker 2015).

Line 87: Mention the effects of body condition score on pregnancy rate.

Response: The authors added in lines 90 to 92: Body condition score is a critical factor influencing nutritional status of beef cows and determining the success of artificial insemination (Carvalho et al., 2022).

Line 99: Which are your hypothesis ? I recommend to avoid using subjective adverbs such as “carefully”. I assume that if your objective is to quantify, you will do it carefully.

Response: The main hypothesis is in lines 106-109. We removed the word “carefully.”

Line 123: Did you considered the changes in breeds genetics during the period evaluated?

Response: No, we did not. The herds were not subjected to formal selection programs, and each herd was only evaluated one or a few years. We surmise that no relevant directional genetic changes took place in the study.

Line 142: Which were the types of supplements used? Was there a range of supplementation levels? 

Response: We included the following table with the information mentioned.

Table 2 – Average and standard error of each variable in the initial database 

Variable Average

Age at the beginning of the breeding season1 24.6±7.5 months

Animal categories

 Heifers

 Primiparous cows 

2257 females

1676 females

Body weigth at the beginning of the breeding season 315.4±55.9 kg

Body weigth at the end of the breeding season 337.3±53.4 kg

Breeds

 Angus

 Braford

 Brangus

 Crossbred

 Devon

 Hereford 

306 females

499 females

323 females

1928 females

110 females

767 females

Body condition score at the beginning of the breeding season2 3.2±0.6

Body condition score at the end of the breeding season2 3.4±0.6

Stocking rate3 337.32±54.68 kg BW/ha

Pasture types

 Cultivated

 Improved pasture4

 Natural grassland 

2050

324

1559

Feed supplementation before the breeding season

 Not supplemented

 Supplemented

 Brown rice bran 

 Commercial concentrate

 Corn grain

 Deffated rice bran 

 Deffated rice bran and sorgum silage

 Ground corn grain

 Sorghum silage and commercial concentrate 

 Protein salt

 Rice and soy bran

 Ryegrass and White clover hay

 Sectaria hay

 Sorghum silage and commercial concentrate 

-

0.5 to 1.0% BW

0.7 to 1.5% BW

0.5% BW

1.5% BW 

1.5% BW

0.7% BW

1.5% BW 

0.1% BW

0.56% BW

0.28% BW

0.92% BW

1.5% BW

Line 187: Maybe the sentence: “This final model…” sould be change with “the final model after simplifications”, since it may generate confusions with the full model. 

Response: The text was changed to:

This final model after simplifications was tested against the full model by a likelihood-ratio test using the anova() function to make sure they were not significantly different.

Line 196: sup.pre was not defined. 

Line 202: sup.breed was not defined. 

Response: We changed added the information about sup.breed. 

The text (lines 201-212) was changed to:

where pasture.pre is a factor indicating whether animals grazed natural grassland, cultivated pastures or improved grassland prior to the breeding period; s.sr is stocking rate, I(s.sr^2) is the quadratic effect of stocking rate, and sup.pre is a binary variable indicating whether animals received supplementation before of the breeding period. Other terms were defined above. Finally, change in body weight during the breeding period (s.dwt) was analyzed starting with the following full model: 

s.dwt ~ categ + breed + pasture.breed + s.sr + sup.breed + I(s.sr^2) + pasture.breed:s.sr + categ:s.sr + breed:s.sr + (1 | experiment)

where pasture.breed is the type of pasture grazed during the breeding period and sup.breed is a binary variable indicating whether animals received supplementation during the breeding period.

Line 205: Is sup.pre defined correctly? Or was it confused with sup.breed? Statistical analyses if you reported de means of body condition score of your database, why didn’t include it in your models? 

Response: The definitions of sup.pre and sup.breed were added as indicated above.

Line 233: “… 440.0 kg or more and had…” delete the “and”. 

Response: Response: The word “and” was excluded. The phrase was changed to:

Pregnancy rate increased steeply with increasing body weight at the beginning of the breeding season for crossbreed and B. taurus females, but for B. taurus females it increased faster and reached a higher maximum than for crossbreed females. B. taurus females starting the breeding season with an average weight of 440.0 kg or more had an expected pregnancy rate of 99.0%, whereas crossbred females that started the breeding season with similar weight had an expected pregnancy rate of 91.0% (Fig 2). 

Line 237: Interaction instead of interactin. 

Response: The phrase was re-written as follows:

Fig. 2. Interaction between body weight at the beginning of the breeding season and Breed on pregnancy rate of heifers and primiparous cows (Sint: crossbred, Taur: B. Taurus females). Shaded strips represent 95% confidence intervals for the expected value. Body 240 mass average and standard deviation were 318 and 69 kg.

Line 282: Instead of “near its average” I would suggest to use range of daily gain.

Response: Actually, the mathematically correct statement should say “at” the average. We changed the sentence to be:

The response to daily gain was similar to that in the model without stocking rate; pregnancy rate increased 1.5% per 100 g of daily gain when daily gain was at its average.

Line 360: Can you discuss this with your results of body condition score? 

Response: We can’t do this because we did not use body condition score in the modeling. We chose to use body weights instead of condition score mainly because of three reasons: (1) body condition score was recorded as a discrete variable with few levels, which prevented its use with traditional gaussian models for quantitative variables, (2) category and body weight together contained most of the information present in body condition score and (3) adding an intermediary “layer” to incorporate the relationships among condition score and the other variables would have complicated the model in a way unnecessary to address our goals. However, we agree that the data could be used for additional exploration of the effects and response of body score, but that would be a different paper.

Line 369: Couldn’t this be corrected by using the body condition score instead of the body weight in the models? 

Response: See response above.

Line 393-396: Couldn’t this be corrected by using the body condition score instead of the body weight in the models? 

Response: See response above.

Line 436: Correct the citation to “in agreement with Rovira (1974)”. 

Response: The citation was changed. The text was changed to: 

In agreement with Rovira (1974), we observed that when heifers and primiparous cows are maintained in optimal conditions of grazing and nutrition, that is, maintained in high quality pastures, with sufficient body weight and intermediate stocking rates, they achieve near maximal reproductive success. Sufficient nutrition allows early breeding, which increases the overall efficiency of production for the herd.

Line 452-458: It’s very interesting the analysis of the effects of stocking rate nor mediated by the effects on BW at breeding and BW gain during breeding. You discussed that this effect occurs only when BW at the beginning of the breeding season is greater than average, and mentioned that parasites could be a possible cause of this response. However, parasites mainly affect weight gain. I suggest to discuss which would be the health and management variables that don’t affect the nutritional condition but do affect the reproductive performance. 

Response: The authors added in the paragraph: However, herd health, particularly related to infections that directly or indirectly compromise the reproductive tract of females and the embryo and/or fetus, also stands out as an important factor of interference in the reproductive efficiency of beef cattle herds. In free herds, the introduction of the etiologic agent will cause, in most cases, various clinical signs such as repeated estrus, abortion, stillbirth, birth of weak animals and infertility (Junqueira & Alfieri, 2006).

Line 484: be used IN herd reproductive management. 

Response: The phrase was changed to:

Body weight at the beginning of the breeding season is an easily measurable variable and can be used as a herd reproductive management tool.

Figure 3: I would suggest to include body mass average and standard deviation as in Figure 2. 

Response: We added the following sentence to the caption:

Body mass average and standard deviation were 318 and 69 kg.

Figure 4 and 5: I would suggest to include Stocking rate average and standard deviation as in Figure 2. 

Response: We added the following sentence to the caption:

Stocking rate average and standard deviation were 328 and 58.9 kg/ha.

Figure 5: The y axis title is incomplete. 

Response: We added the missing closing parenthesis.

Figure 6: This is a great representation, however the size of the figures is too small and it’s difficult to read. The references to the type of line are not clear, and it is hard to distinguish the thick of the lines. 

Response: We created a new figure where the individual graphs are larger.

The authors

---

## [Decision Letter · Decision Letter 2]

19 Sep 2022

Direct and indirect nutritional factors that determine reproductive performance of heifer and primiparous cows1

PONE-D-21-30153R2

Dear Dr. Eloy,

We’re pleased to inform you that your manuscript has been judged scientifically suitable for publication and will be formally accepted for publication once it meets all outstanding technical requirements.

Kind regards,

Luis Alonso Villalobos

Academic Editor

PLOS ONE

Additional Editor Comments (optional):

Dear author.

It is my pleasure to inform that your manuscript has been accepted for publication in PLOS ONE. We appreciate the timely manner in which you made the edits suggested by the reviewers as well as your patience while we received the comments from the reviewers.

Reviewers' comments:

Reviewer's Responses to Questions

**Comments to the Author**

1. If the authors have adequately addressed your comments raised in a previous round of review and you feel that this manuscript is now acceptable for publication, you may indicate that here to bypass the “Comments to the Author” section, enter your conflict of interest statement in the “Confidential to Editor” section, and submit your "Accept" recommendation.

Reviewer #3: (No Response)

Reviewer #5: All comments have been addressed

2. Is the manuscript technically sound, and do the data support the conclusions?

Reviewer #3: Yes

Reviewer #5: Yes

3. Has the statistical analysis been performed appropriately and rigorously? 

Reviewer #3: Yes

Reviewer #5: Yes

4. Have the authors made all data underlying the findings in their manuscript fully available?

Reviewer #3: Yes

Reviewer #5: Yes

5. Is the manuscript presented in an intelligible fashion and written in standard English?

Reviewer #3: Yes

Reviewer #5: Yes

6. Review Comments to the Author

Reviewer #3: NO FURTHER COMMENTS

Reviewer #5: (No Response)

7. PLOS authors have the option to publish the peer review history of their article (what does this mean?). If published, this will include your full peer review and any attached files.

Reviewer #3: No

Reviewer #5: No

---

## [Editor Report · Acceptance letter]

22 Sep 2022

PONE-D-21-30153R2 

Direct and indirect nutritional factors that determine reproductive performance of heifer and primiparous cows1 

Dear Dr. Eloy:

I'm pleased to inform you that your manuscript has been deemed suitable for publication in PLOS ONE. Congratulations! Your manuscript is now with our production department. 

Kind regards, 

on behalf of

Dr. Luis Alonso Villalobos 

Academic Editor

PLOS ONE